# Defective mitochondrial DNA homeostasis in the substantia nigra in Parkinson disease

Christian Dölle[1,2], Irene Flønes[1,2], Gonzalo S. Nido[1,2], Hrvoje Miletic[3,4], Nelson Osuagwu[1,2], Stine Kristoffersen[3,5], Peer K. Lilleng[3,5], Jan Petter Larsen[6], Ole-Bjørn Tysnes[1,2], Kristoffer Haugarvoll[1,2], Laurence A. Bindoff[1,2] & Charalampos Tzoulis[1,2]

Increased somatic mitochondrial DNA (mtDNA) mutagenesis causes premature aging in mice, and mtDNA damage accumulates in the human brain with aging and neurodegenerative disorders such as Parkinson disease (PD). Here, we study the complete spectrum of mtDNA changes, including deletions, copy-number variation and point mutations, in single neurons from the dopaminergic substantia nigra and other brain areas of individuals with Parkinson disease and neurologically healthy controls. We show that in dopaminergic substantia nigra neurons of healthy individuals, mtDNA copy number increases with age, maintaining the pool of wild-type mtDNA population in spite of accumulating deletions. This upregulation fails to occur in individuals with Parkinson disease, however, resulting in depletion of the wild-type mtDNA population. By contrast, neuronal mtDNA point mutational load is not increased in Parkinson disease. Our findings suggest that dysregulation of mtDNA homeostasis is a key process in the pathogenesis of neuronal loss in Parkinson disease.

[1] Department of Neurology, Haukeland University Hospital, 5021 Bergen, Norway. [2] Department of Clinical Medicine, University of Bergen, 5020 Bergen, Norway. [3] Department of Pathology, Haukeland University Hospital, 5021 Bergen, Norway. [4] Department of Biomedicine, University of Bergen, 5020 Bergen, Norway. [5] Gade Laboratory for Pathology, Department of Clinical Medicine, Haukeland University Hospital and University of Bergen, 5021 Bergen, Norway. [6] Network for Medical Sciences, University of Stavanger, 4036 Stavanger, Norway. Correspondence and requests for materials should be addressed to C.T. (email: charalampos.tzoulis@nevro.uib.no).

Somatic mitochondrial DNA (mtDNA) damage has been associated with both normal aging and neurodegeneration[1]. Accelerated mtDNA mutagenesis causes a premature aging phenotype in mice[2] and somatic mtDNA deletions have been shown to accumulate with advancing age in post-mitotic tissues including the brain, heart and skeletal muscle[3]. In the brain, the dopaminergic substantia nigra is particularly susceptible to somatic mtDNA deletions, which accumulate there at substantially higher levels compared with other brainstem nuclei, deep grey structures, or the cerebral and cerebellar cortex. This predilection has led to the hypothesis that mtDNA damage plays a role in the pathogenesis of Parkinson disease (PD)[4,5], where neurodegeneration of the substantia nigra is the main pathological hallmark and widely accepted as the cause of the cardinal clinical features[6]. mtDNA deletions were shown to accumulate at similar levels in both individuals with PD and healthy controls however[4], and therefore do not provide a sufficient explanation for the specific vulnerability of the substantia nigra in PD. Mice accumulating high levels of mtDNA deletion due to an error-prone mtDNA-polymerase (POLG) show a concomitant increase in mtDNA copy number which is associated with nigrostriatal survival and even resistance to mitochondrial respiratory chain complex-I inhibition[7]. Although a similar protective mechanism has not yet been identified in humans, the importance of mtDNA copy-number regulation is highlighted by the vulnerability of the substantia nigra in inherited mtDNA-depletion disorders[8,9] and the increased risk of PD associated with genetic variation in genes encoding key factors of mtDNA maintenance, such as the mtDNA polymerase γ (POLG) and mitochondrial transcription factor A (TFAM)[10,11]. Nevertheless, the precise mechanism by which mtDNA copy-number loss contributes to brain aging and neurodegeneration remains unclear.

We hypothesized that dopaminergic substantia nigra neurons in PD are rendered vulnerable to the effects of age-dependent mitochondrial mutagenesis due to an underlying dysregulation of mtDNA homeostasis. To test our hypothesis, we employed an integrative approach to study the complete spectrum of mtDNA changes in individual neurons from individuals with PD and controls.

Our sample was derived from a population-based, prospectively collected and extensively characterized cohort[12]. To ensure our sample was homogenous and representative for sporadic PD, we excluded known monogenic causes of PD by whole-exome sequencing. Neuropathological examination confirmed Lewy-body disease in all PD samples, whereas control samples were negative for neurodegenerative markers (Supplementary Table 1).

Our results show that dopaminergic substantia nigra neurons of individuals with PD accumulate higher levels of somatic mtDNA deletions, but not point mutations, compared with age-matched controls. Moreover, in healthy individuals, mtDNA copy number increases with age, thus maintaining the pool of wild-type mtDNA population in spite of accumulating deletions. Conversely, mtDNA copy number does not increase in individuals with PD, resulting in depletion of the wild-type mtDNA population. Our findings suggest that mtDNA homeostasis is impaired in the substantia nigra of individuals with PD.

## Results

**mtDNA copy number upregulation in aging substantia nigra.** First we characterized age-dependent mtDNA changes in different neuronal populations from 21 controls aged 11–87 years (Supplementary Table 1). We studied single dopaminergic neurons from the ventrolateral tier (area A9) of the substantia nigra pars compacta, pyramidal neurons from the frontal cortex and Purkinje cells of the cerebellar cortex (Fig. 1). The proportion of mtDNA molecules harbouring major arc deletions showed a significant positive correlation with age in nigral neurons ($r = 0.39$, $P = 1 \times 10^{-6}$; Fig. 2a), which is in line with previous reports[4,5]. Total mtDNA copy number also increased with age ($r = 0.29$, $P = 4 \times 10^{-4}$; Fig. 2b) and, strikingly, showed a significant positive correlation with the level of deletion in each neuron ($r = 0.40$, $P = 2 \times 10^{-6}$; Fig. 2c). Due to this concomitant copy number increase, absolute levels of wild-type (non-deleted) mtDNA did not decrease over time ($r = 0.03$, $P = 0.3$) in spite of

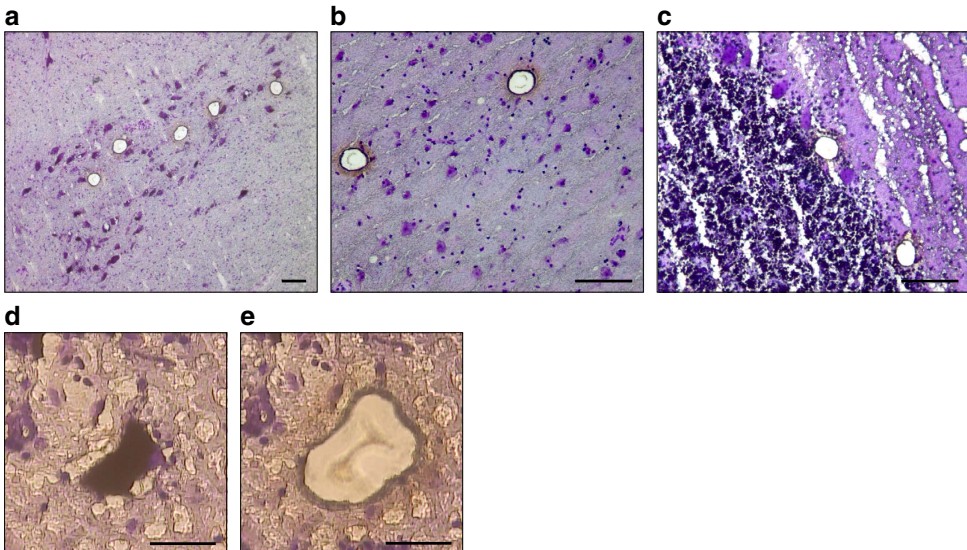

**Figure 1 | Laser microdissection (LMD) and workflow of mtDNA analyses in single neurons.** (**a–c**) Representative examples of thick (20 µm) sections of fresh-frozen brain tissue stained with cresyl violet and used for LMD. (**a**) Pigmented, dopaminergic neurons were picked from the substantia nigra pars compacta ( × 100). (**b**) Pyramidal neurons were collected from the prefrontal cortex ( × 200). (**c**) Purkinje cells where picked from the cerebellar cortex ( × 200). Oval holes in the tissue show where neurons have been microdissected and collected for analysis. (**d,e**) Microphotographs showing a dopaminergic neuron of the substantia nigra before (**d**) and after (**e**) microdissection ( × 400). Each neuron was collected individually in a 0.5 ml reaction tube containing 15 µl lysis buffer, lysed overnight and subjected to PCR analysis. Scale bars, 100 µm (**a–c**) and 20 µm (**d,e**).

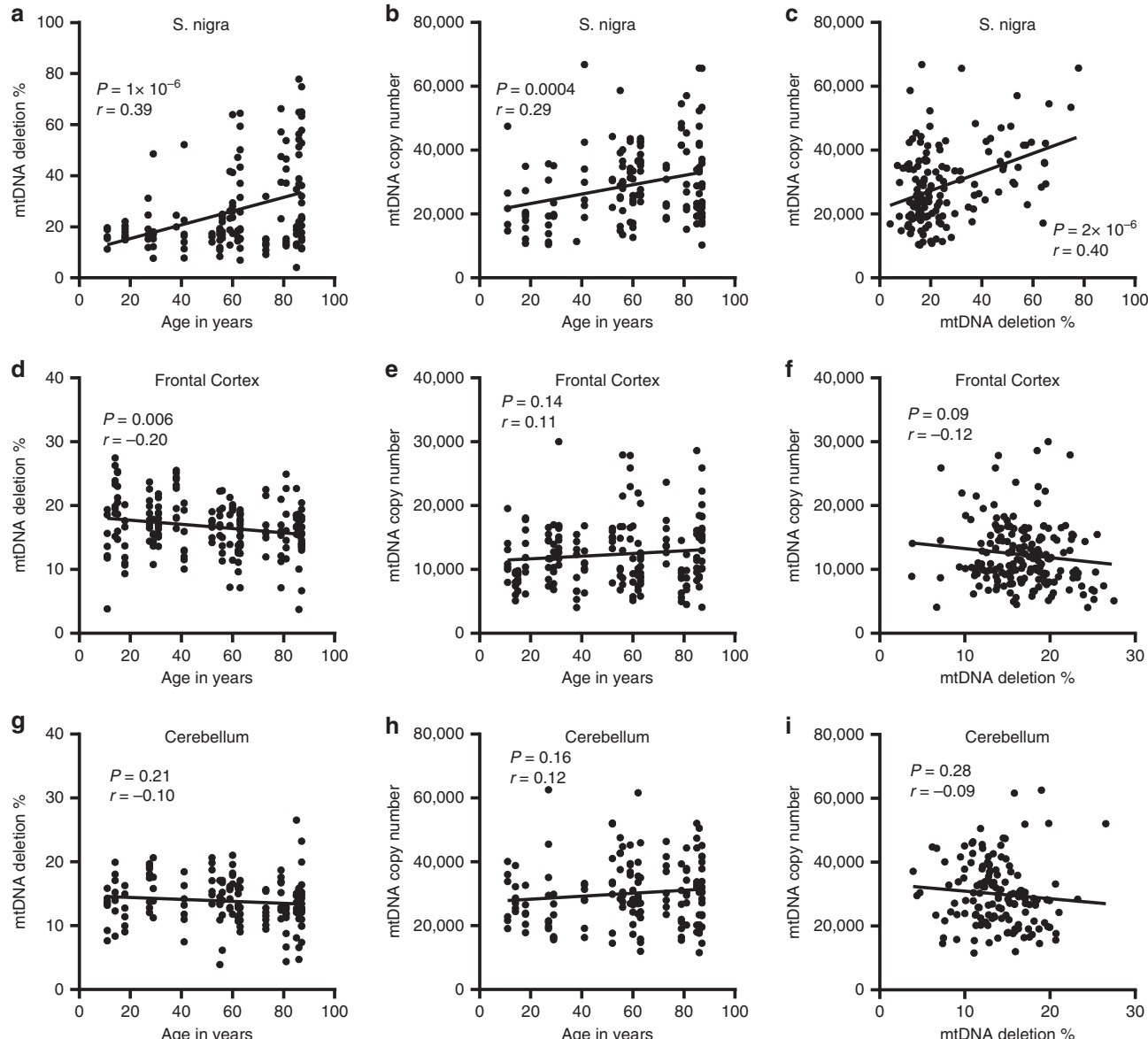

**Figure 2 | Age-dependent mtDNA changes in three neuronal populations from neurologically healthy controls.** Single neurons from three brain areas of neurologically healthy controls ($n = 21$) were analysed for mtDNA copy number and deletion level.(**a–c**) Single dopaminergic neurons of the pars compacta of the substantia nigra ($n = 147$). (**d–f**) Pyramidal neurons of the frontal cortex ($n = 193$). (**g–i**) Purkinje cells of the cerebellum ($n = 148$). All three areas were analysed from all subjects ($n = 21$). (**a**) In the substantia nigra, the fraction (%) of mtDNA harbouring major arc deletions increases significantly with age ($n = 147$, $P = 1 \times 10^{-6}$; Pearson correlation). This is accompanied by a concomitant increase of mtDNA copy number (**b**), which correlates strongly with the levels of mtDNA deletion in each neuron ($n = 147$, $P = 2 \times 10^{-6}$; Pearson correlation) (**c**). Frontal neurons show no increase in mtDNA deletion levels ($n = 193$, $r = -0.20$, $P = 0.06$; Pearson correlation) (**d**) or copy number ($n = 193$, $r = 0.11$, $P = 0.14$; Pearson correlation) (**e**) with age and no correlation between mtDNA copy number and levels of mtDNA deletion ($n = 193$, $r = -0.12$, $P = 0.09$; Pearson correlation) (**f**). Similarly, Purkinje cells of the cerebellum show no increase in mtDNA deletion levels ($n = 148$, $r = -0.10$, $P = 0.21$; Pearson correlation) (**g**) or copy number ($n = 148$, $r = 0.12$, $P = 0.16$; Pearson correlation) (**h**) with age and mtDNA copy number does not correlate with the levels of mtDNA deletion ($n = 148$, $r = -0.09$, $P = 0.28$; Pearson correlation) (**i**). Each dot represents data from a single neuron. Statistics are derived from Pearson correlation and linear regression analyses.

progressively increasing proportion of deletion. Whereas both deletion and age appeared to correlate with mtDNA copy number, using a multiple linear regression model, we found that only deletion was a statistically significant predictor of copy number (model: $r = 0.40$, $R^2 = 0.18$, $P = 6.3 \times 10^{-7}$; deletion: $\beta = 0.34$, $P = 6 \times 10^{-5}$; age: $\beta = 0.16$, $P = 0.06$).

In frontal neurons and Purkinje cells, the level of mtDNA deletion was generally low and did not increase with age (Fig. 2d,g). Similarly, total mtDNA copy number showed no significant change with age (Fig. 2e,h) or correlation with the levels of deletion in these cells (Fig. 2f,i). Thus, our findings

demonstrate that aging substantia nigra neurons can robustly upregulate mtDNA copy number and this correlates with the levels of somatic mtDNA deletion in each cell. We suggest that this reflects an intrinsic neuroprotective mechanism allowing neurons to maintain an adequate pool of wild-type (non-deleted) mtDNA molecules despite the age-dependent accumulation of somatic deletion.

**Evidence of impaired mtDNA copy-number regulation in PD.** We next investigated the integrity of neuronal mtDNA homeostasis in single neurons from individuals with validated sporadic

PD ($n = 10$) compared with neurologically healthy age- and sex-matched controls ($n = 10$; Supplementary Table 1). Dopaminergic nigral neurons from individuals with PD contained significantly higher levels of mtDNA deletions than nigral neurons from healthy controls (PD mean $40.2 \pm 20.0\%$, controls mean $31.5 \pm 19.5\%$, $P = 0.004$; Fig. 3a). Notably, the proportion of neurons with deletion levels exceeding 60%, which has been postulated as a threshold for respiratory dysfunction in cells[13], was twice as high in the PD group (21.4%, $n = 84$) compared with controls (10.8%, $n = 74$).

In spite of higher levels of deletion, mean total neuronal mtDNA content was similar in the two groups (PD mean $31,271 \pm 13,217$, controls mean $32,148 \pm 11,970$ copies/neuron, $P = 0.57$; Fig. 3b, total). Correlation of mtDNA copy number with the age of the individual or mtDNA deletion levels in each neuron revealed, however, striking differences between PD and controls. In line with our initial findings, mtDNA copy number increased significantly with age in the group of age-matched controls and correlated with the level of deletion ($r = 0.50$, $P = 5 \times 10^{-5}$; Fig. 3c), whereas in the PD group, the rate of copy number

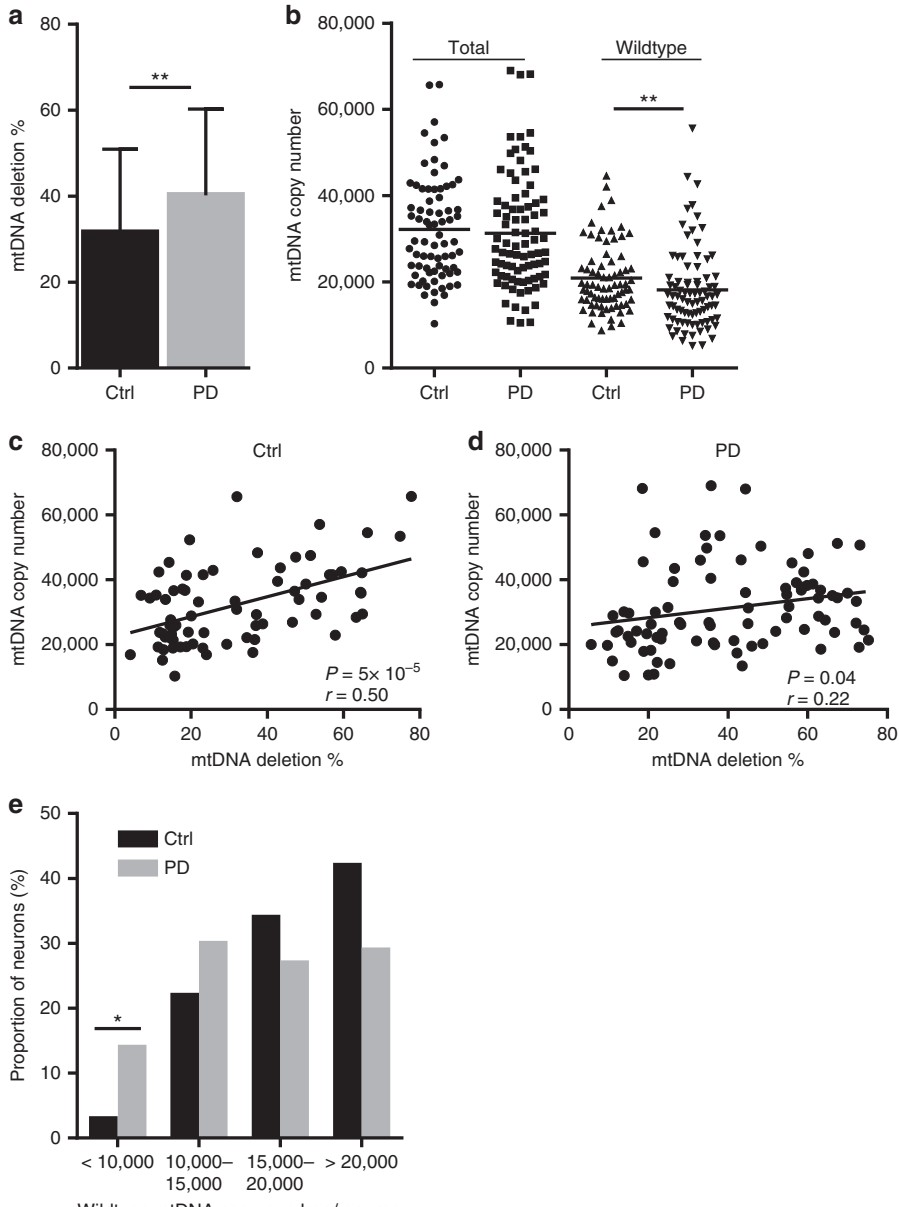

**Figure 3 | Impaired mtDNA maintenance in the dopaminergic substantia nigra of patients with PD.** Data show analyses in single dopaminergic substantia nigra neurons from individuals with PD ($n = 84$ neurons from 10 individuals) and age-matched controls ($n = 74$ neurons from 10 individuals). (**a**) Deletion levels are significantly higher in PD compared with controls ($P = 0.004$; Mann–Whitney $U$ test). Error bars show s.d. (**b**) Scatter plot of total and wild-type (non-deleted) mtDNA. Total mtDNA copy number is similar in PD and controls, but the levels of wild-type mtDNA are significantly decreased in PD ($P = 0.006$; Mann–Whitney $U$ test). Bars show mean. (**c,d**) Linear regression of neuronal mtDNA copy number plotted against deletion levels; each point shows data from a single neuron. The correlation between mtDNA copy number and deletion is poor in PD ($P = 0.04$, $r = 0.22$) compared with controls ($P = 5 \times 10^{-5}$, $r = 0.50$; Pearson correlation). (**e**) Fraction of total dopaminergic substantia nigra neurons of individuals with PD and controls distributed according to wild-type (non-deleted) mtDNA content. The fraction of neurons with very low ($<10,000$) wild-type mtDNA copy number is significantly enriched in PD (Ctrl 2.7% versus PD 14.3%, $P = 0.01$; Fisher's exact test), whereas in controls, the majority of neurons contain at least 15,000 copies of non-deleted mtDNA. Ctrl: controls. *$P < 0.05$, **$P < 0.01$.

increase was substantially less pronounced ($r = 0.22$, $P = 0.04$; Fig. 3d) and, in the presence of higher levels of deletion, resulted in quantitative loss of wild-type mtDNA (PD mean 18,109 ± 9,421, controls mean 20,876 ± 7,839 copies/neuron, $P = 0.0056$; Fig. 3b, wild type). Although the degree of wild-type mtDNA loss appeared mild at the group level, substantial variation was seen between individual cells. Notably, the subset of neurons with severe wild-type depletion, defined as <10,000 copies per neuron, was significantly higher in the PD group (PD 14.3%, controls 2.7%, $P = 0.01$; Fig. 3e). This corresponds to a loss of ~70% of the mean wild-type mtDNA population in these neurons, which is higher than the estimated threshold for respiratory dysfunction[13].

mtDNA deletion levels were generally low in frontal neurons (PD mean 15.50 ± 3.75%, controls mean 15.57 ± 3.90%, $P = 0.1$) and Purkinje cells (PD mean 13.07 ± 3.90%, controls mean 13.04 ± 3.70%, $P = 0.8$) of both individuals with PD and controls (Fig. 4a,b). mtDNA copy number in these cells showed subtle differences between the groups (Fig. 4c,d), but there was no correlation with the levels of deletion (Fig. 4e–h).

**No evidence of impaired mitochondrial biogenesis in PD**. To investigate whether the loss of wild-type mtDNA in the PD group reflects a defect of mtDNA maintenance specifically, or an overall defect in mitochondrial biogenesis, we investigated markers of mitochondrial mass and biogenesis in the substantia nigra by immunohistochemistry *in situ*, in brain sections from individuals with PD and healthy controls. Total mitochondrial mass was assessed using the mitochondrial membrane marker porin, and mitochondrial biogenesis signalling was assessed via the peroxisome proliferator-activated receptor gamma co-activator 1-alpha (PGC-1α), a transcription co-activator that regulates mitochondrial biogenesis. Morphometric analyses of the sections revealed no detectable difference in either intensity or cellular distribution of the porin or PGC-1α staining between individuals with PD and controls, suggesting no overt change in mitochondrial biogenesis signalling or total mass in dopaminergic neurons (Figs 5 and 6).

**mtDNA somatic point mutations are not elevated in PD**. To assess whether point mutations also contribute to neuronal mtDNA pathology in PD, we performed ultra-deep sequencing of

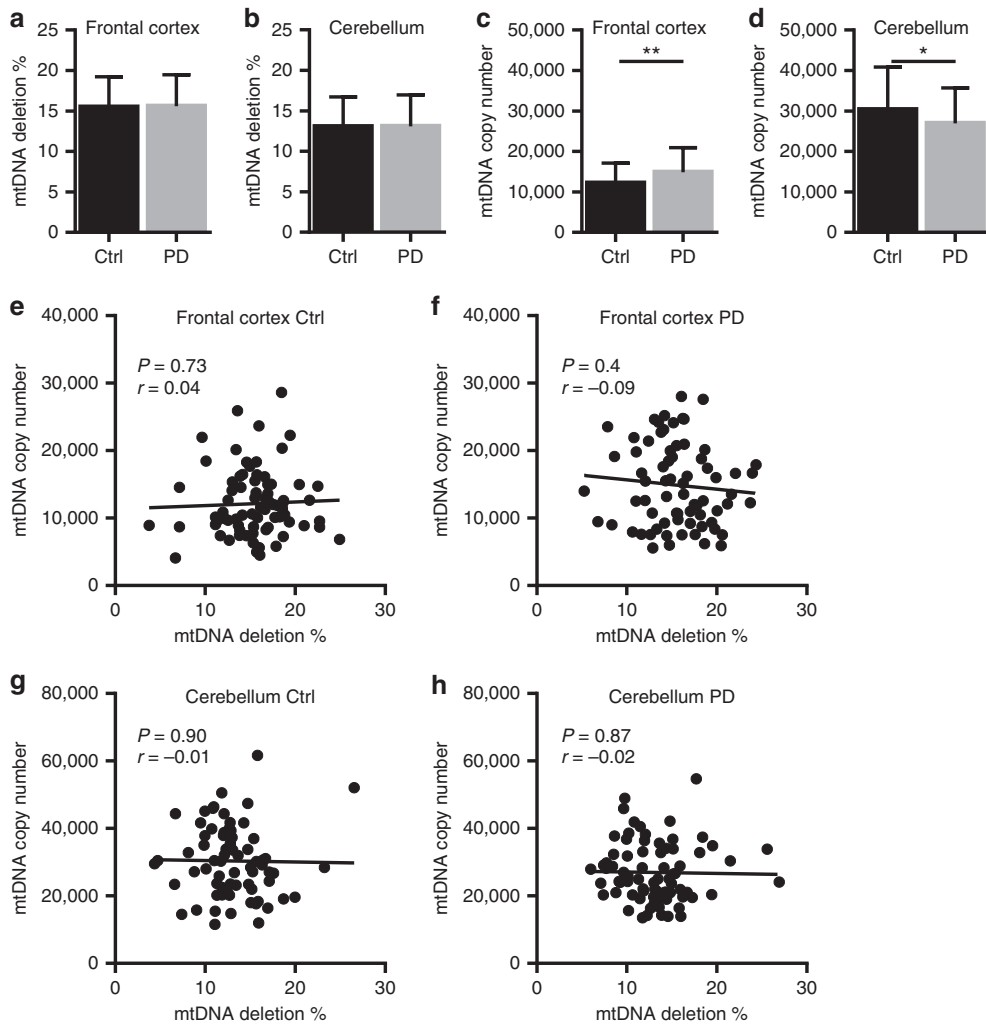

**Figure 4 | mtDNA analyses in frontal and cerebellar neurons.** mtDNA analyses in single pyramidal neurons from frontal cortex and Purkinje cells of the cerebellum of individuals with PD ($n = 73$ frontal neurons and 76 Purkinje cells from 10 individuals) and healthy controls ($n = 78$ frontal neurons and 71 Purkinje cells from 10 individuals). (**a,b**) mtDNA deletion levels. (**c,d**) mtDNA copy number. Error bars show s.d. Comparison by Mann–Whitney $U$ test. *$P = 0.04$, **$P = 0.005$. (**e–h**) Correlation between mtDNA copy number and deletion levels in frontal cortex (Ctrl: $r = 0.04$, $P = 0.73$; PD: $r = -0.09$, $P = 0.4$; Pearson correlation) (**e,f**) and cerebellum (Ctrl: $r = -0.01$, $P = 0.90$; PD: $r = -0.02$, $P = 0.87$; Pearson correlation) (**g,h**) from individuals with PD (**f,h**) and controls (**e,g**). Ctrl: controls.

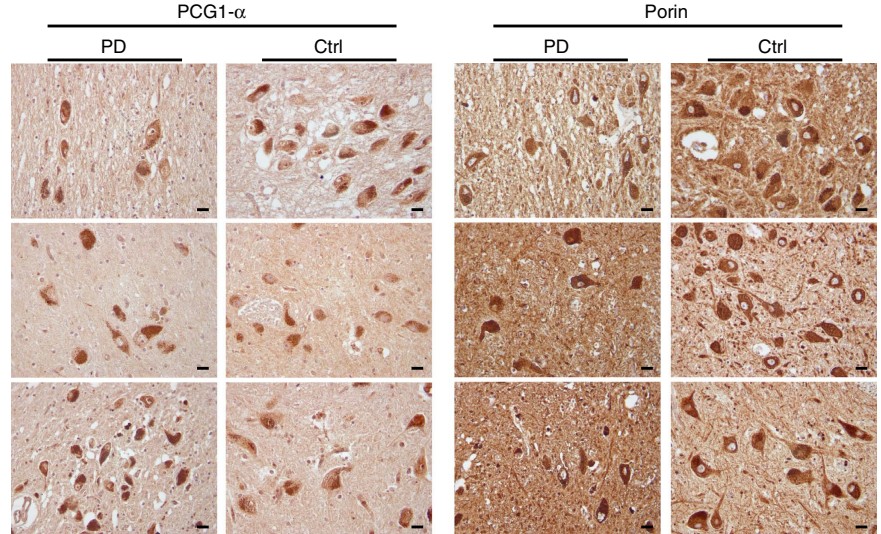

**Figure 5 | Immunohistochemistry for porin and PGC-1α in dopaminergic substantia nigra neurons of individuals with PD and controls.** Representative sections are shown from three individuals with PD and three controls (Ctrl). There is no detectable difference in staining intensity or distribution between individuals with PD and controls. All pictures have been taken at ×400 magnification. Scale bars, 20 μm.

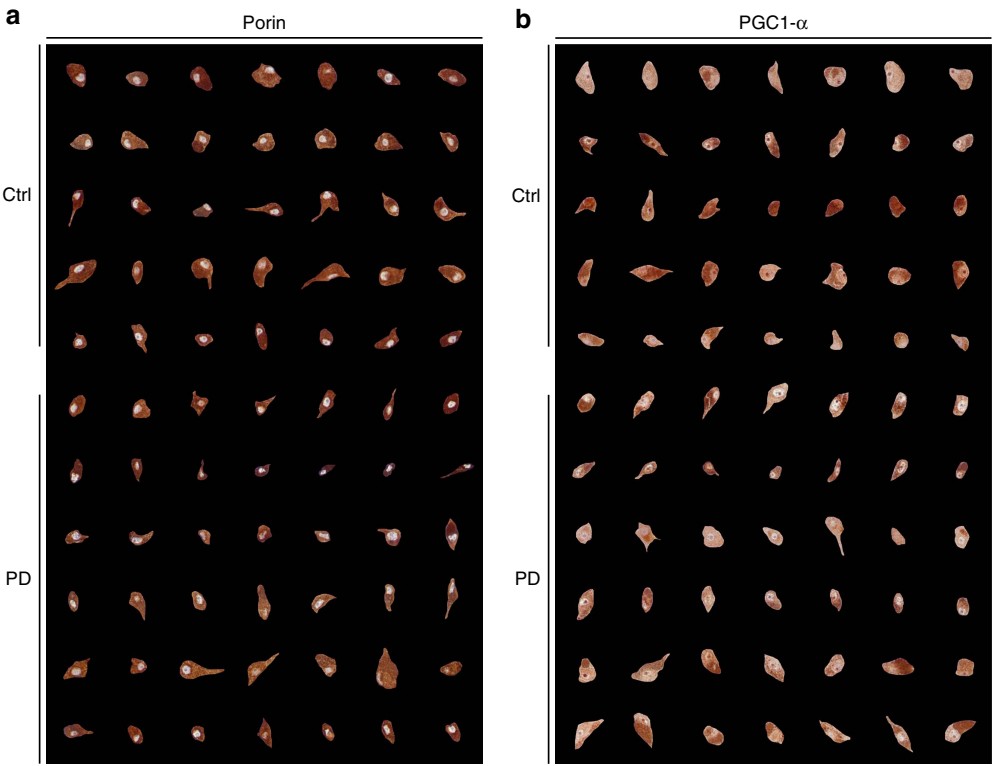

**Figure 6 | Single-cell montage of porin and PCG-1α-stained dopaminergic substantia nigra neurons of individuals with PD and controls.** The photomontages show representative examples of individual dopaminergic neurons from the ventrolateral tier of the substantia nigra pars compacta, stained with antibodies against porin (**a**) and PGC-1α (**b**). Each row shows neurons from the same individual. In both panels, the top five rows are controls and bottom six rows individuals with PD. Magnification: ×400.

a ∼5 kb fragment of mtDNA (rCRS 1157–5924) in a total of 184 single dopaminergic substantia nigra neurons from the 10 individuals with PD and 10 matched controls. The sequenced fragment was localized away from the deletion region to ensure it represented the total mtDNA population of each neuron (for details see the 'Methods' section and Fig. 7a,b). Sequence data from 144 neurons successfully passed quality control and were used in the analyses. High-sequence coverage (Fig. 7c) allowed for

detection of single-nucleotide variants (SNVs) as low as at a heteroplasmic frequency (HF) of 0.1% (Fig. 7d). The point mutation frequency was estimated as number of SNVs per 1,000 bp mtDNA.

Homoplasmic (that is, affecting all mtDNA molecules) SNVs (HF > 98%) had a mean frequency of $0.95 \pm 0.47/1,000$ bp per neuron and were shared by all neurons of an individual, consistent with being inherited. The number of transitions

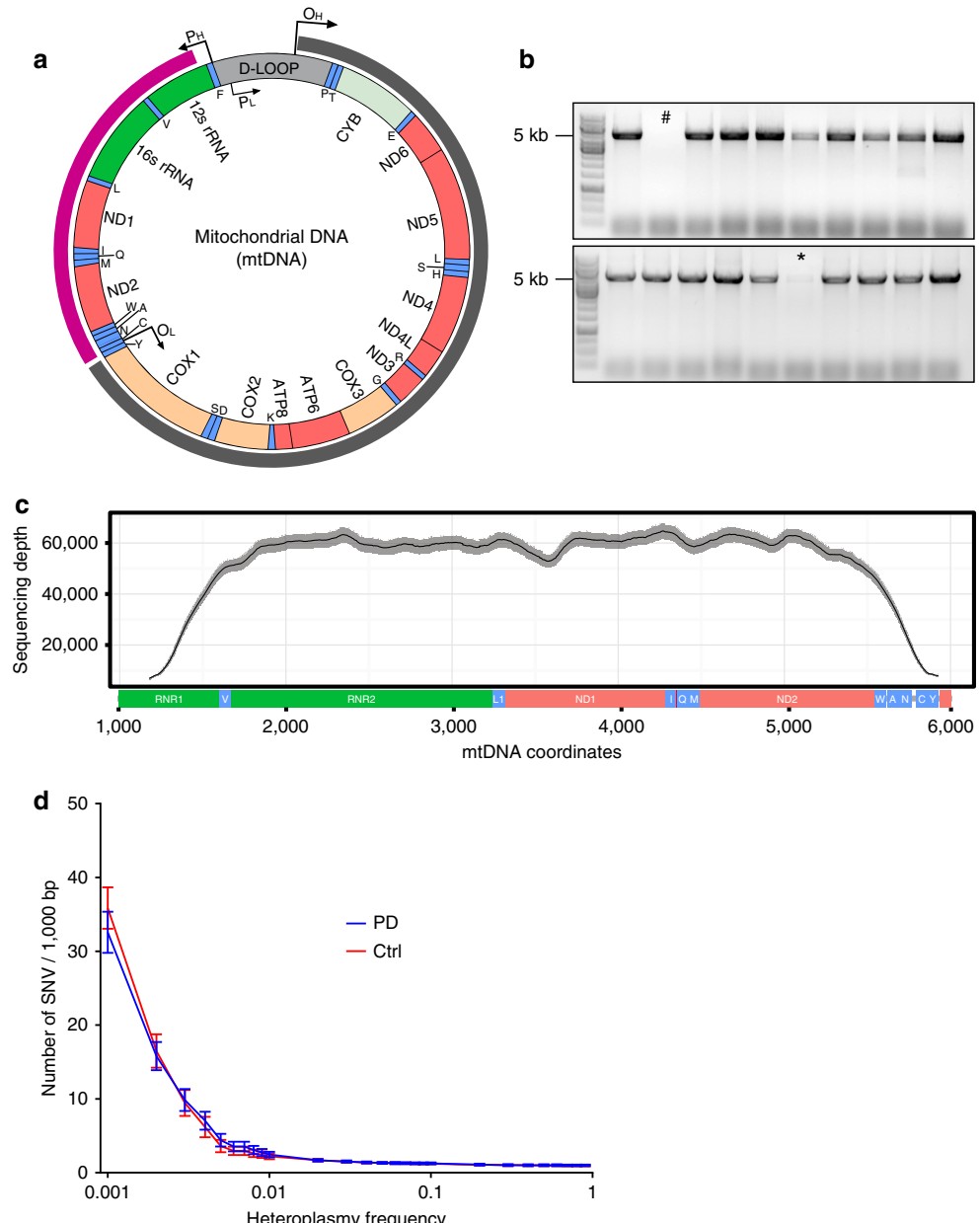

**Figure 7 | Amplification and deep-sequencing of mtDNA from single dopaminergic substantia nigra neurons of individuals with PD and controls.**
(**a**) Schematic representation of human mtDNA, indicating the major arc (grey arc), where the vast majority of mtDNA deletions occur, and the position of the 4,767 bp fragment used for ultra-deep sequencing (magenta arc). mtDNA genes are coloured by type of transcript and designated by standard nomenclature. Genes encoding tRNAs are coloured green and designated by the one-letter code of their corresponding amino acid. (**b**) PCR products were amplified from single-cell lysates and analysed by agarose gel electrophoresis before sequencing. Samples with low (*) or no (#) amplification were omitted. (**c**) Mean sequencing depth and coverage of the mtDNA fragment after completed quality control. The solid black line shows the mean depth for all samples per mtDNA position and the shaded grey line corresponds to the 95% confidence interval (CI). Indicatively, 95% of the sequence was covered at $>12,000\times$ depth and 80% at $>50,000\times$ depth. (**d**) Total burden of mtDNA SNVs in single dopaminergic neurons of the substantia nigra plotted against the base 10 logarithm of heteroplasmy frequency (HF). Bars show 95% CIs. Point mtDNA variation is found at generally low levels in nigral neurons and the majority of variants cluster at heteroplasmic frequencies below 1%. There is no difference in point mutational burden between individuals with PD (blue) and controls (red).

(exchange between two purine or pyrimidine bases; $0.92 \pm 0.46/1,000$ bp) greatly outnumbered that of transversions (exchange between a purine and a pyrimidine base; $0.03 \pm 0.07/1,000$ bp), in line with the high transition bias of mammalian mtDNA[14]. The mean load of heteroplasmic (that is, affecting a fraction of mtDNA molecules) SNVs (HF: 0.1–98%) was $33.14 \pm 10.03/1,000$ bp per neuron and most of these ($31.68 \pm 10.96$) clustered at the low-frequency spectrum

(HF 0.1–1%) in both PD and control groups (Fig. 7d). The vast majority (98.8%) of the heteroplasmic changes were not shared across the neurons of an individual, suggesting that these were indeed true somatic changes. Heteroplasmic changes occurred in all sequenced mtDNA genes, with small inter-genic differences (Supplementary Table 3).

The overall burden of heteroplasmic SNVs was similar in individuals with PD and controls (PD $31.63 \pm 10.31$, controls

34.82 ± 9.55 variants per neuron, $P = 0.8$) across the range of HF (Fig. 8a,b). The physical distribution of variation showed no significant gene-specific, or other regional difference between PD and controls (Fig. 8c and Supplementary Table 3). The transition to transversion ratio (Ti/Tv) of heteroplasmic SNVs was similarly low in both groups (PD 2.27 ± 1.63, controls 2.03 ± 1.48, $P = 1$) implying a high fraction of random mutation which resonated with the high proportion of somatic mutation among the heteroplasmic changes (Fig. 8d). The proportion of G:C to T:A transversions which are commonly associated with oxidative DNA damage[15] was also similar in the two groups (PD 0.28 ± 0.08, controls 0.32 ± 0.09, $P = 0.6$), implying that

oxidative stress is not a major factor in somatic mtDNA mutagenesis in PD (Fig. 8e).

## Discussion

In this work, we characterize the complete spectrum of neuronal mtDNA changes in aging and PD, and measure the range of mtDNA damage in the same neuron. We show that deletions and copy-number regulation, but not point mutations, are the main determinants of somatic, neuronal mtDNA damage in humans. Moreover, we show that dynamic regulation of mtDNA copy number occurs in aging substantia nigra neurons allowing them to maintain their pool of wild-type mtDNA in spite of

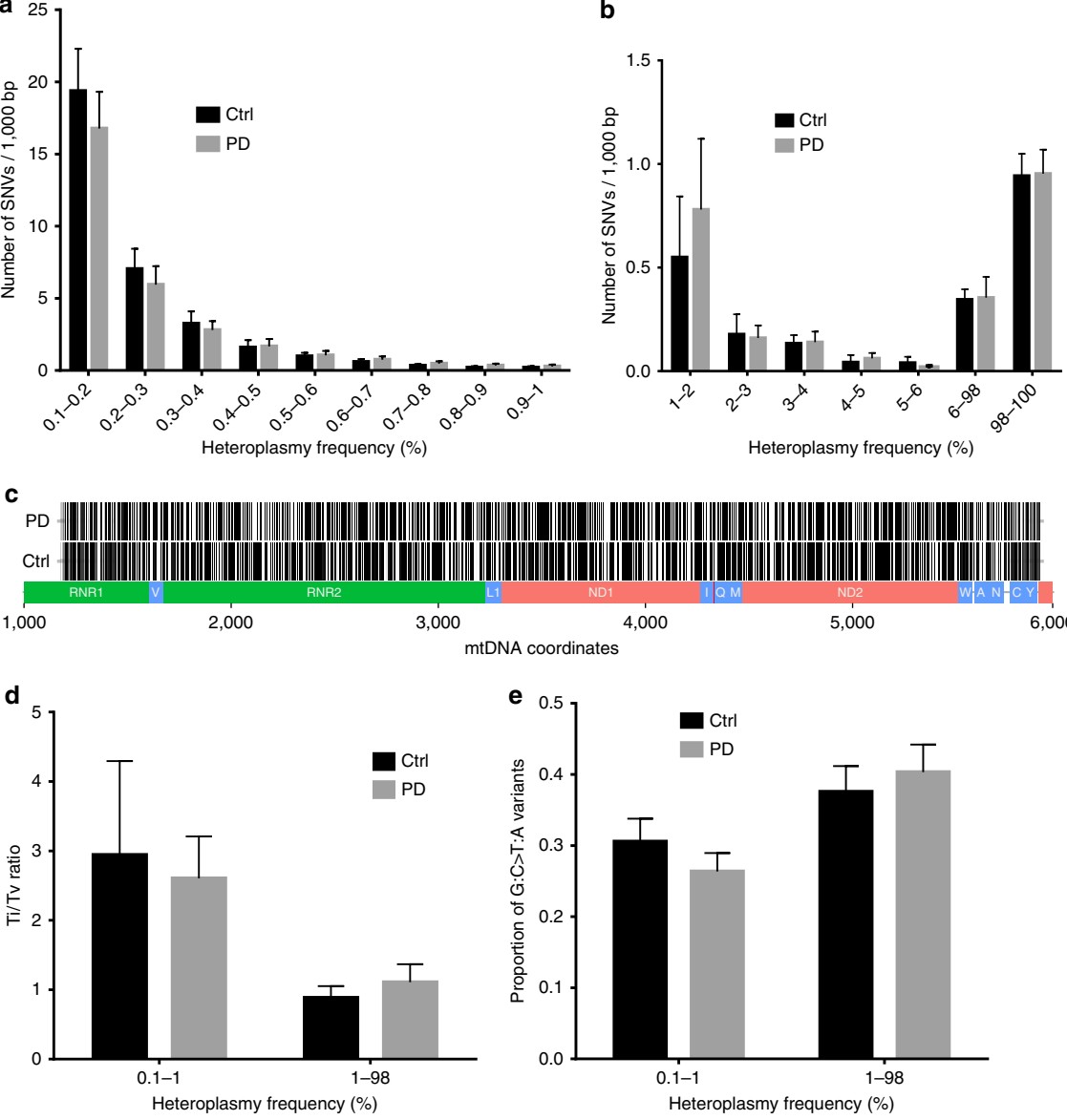

**Figure 8 | Ultra-deep mtDNA sequencing in single dopaminergic neurons of the substantia nigra.** (**a,b**) Burden of mtDNA SNVs per neuron plotted against heteroplasmy frequency (HF). mtDNA point mutational burden is similar in individuals with PD and controls across the range of HF. Error bars show 95% confidence intervals (CIs). (**c–e**) Distribution and type of mtDNA SNVs in neurons of individuals with PD and controls. (**c**) Physical distribution of low-frequency SNVs (0.1–1%) on the sequenced mtDNA fragment in individuals with PD and controls. Each variant is represented by a vertical black streak to allow differentiation between areas with low mutation frequencies (lighter shade) to high-mutation frequencies (darker shade). mtDNA positions and physical gene location are shown in scale on the x axis. There is no significant difference in regional SNVs distribution between regions/genes or between individuals with PD and controls. (**d**) Ratio of mtDNA transitions to transversions (Ti/Tv) at low (0.001–0.01) and high (0.01–0.98) heteroplasmic frequencies. PD and controls show no difference in transition/transversion composition across heteroplasmic frequencies ($\chi^2$). (**e**) The proportion of G:C > T:A tranversions is similar in individuals with PD and controls. Error bars show 95% CIs.

accumulating somatic deletions, similar to what was shown in the POLG mutator mouse[7]. Strikingly, this compensation appears to be blunted in the context of PD resulting in depletion of the wild-type mtDNA pool. We believe this may contribute to the respiratory deficiency observed in these neurons[16], thus playing an important role in the cascade of neurodegeneration.

The mechanism underlying the apparent impairment of mtDNA upregulation in PD could reflect a defect of overall mitochondrial biogenesis or mtDNA homeostasis specifically. Our immunohistochemistry experiments in the substantia nigra detected no measurable difference in the neuronal protein levels of mitochondrial marker porin, or master activator of mitochondrial biogenesis PGC-1α. One limitation of these experiments is that immunohistochemistry has limited sensitivity and would not detect subtle changes of expression. Nevertheless, these results suggest that there is no major impairment in overall mitochondrial biogenesis in PD. It is therefore more likely that the observed lack of mtDNA increase we found in PD reflects a defect in mtDNA homeostasis. This is in line with inherited disorders of mtDNA maintenance due to POLG or helicase Twinkle mutations, where the combination of mtDNA depletion and deletion without an overall change in mitochondrial mass leads to respiratory failure and severe substantia nigra degeneration[8,9]. Impaired mtDNA maintenance has indeed been shown to be particularly deleterious for the substantia nigra and more likely to cause nigrostriatal degeneration compared with other mitochondrial defects[17].

mtDNA maintenance is mediated via a complex interaction of several molecular processes including mtDNA replication and repair, mitochondrial nucleotide pool maintenance, mitochondrial dynamics and quality control[18,19]. It is possible that impairment of one or more of these processes occurs in PD due to genetic variation and/or epigenetically mediated altered transcriptional regulation of the factors involved. Further research is warranted to uncover the mechanisms underlying abnormal mtDNA maintenance in the substantia nigra of individuals with PD.

Our findings appear to contradict recent reports of total mtDNA depletion[20] and higher point mutational load[21] in the brain of individuals with PD. Since those studies were performed in tissue homogenate and not individual neurons, however, they are likely to be confounded by the substantial neuronal loss and gliosis in the PD samples, rather than reflect true changes in dopaminergic neurons. It has been shown that accurate determination of mtDNA quantity in neurodegeneration requires experiments in isolated neurons, and that measurements in homogenate can lead to substantially over- or underestimation of mtDNA quantity depending on underlying tissue pathology[22].

Another important advantage of our study over previous experiments is the ultra-deep coverage of our sequencing experiments. Our high sequencing depth allowed the detection of variants down to a frequency of 0.1%, which, with an average of ∼30,000 mtDNA molecules per neuron, corresponds to as few as 30 mtDNA molecules. Thus we were able to confidently assess point mutations at heteroplasmic frequencies below 1%, where we show that the vast majority (96%) of somatic mtDNA point mutations occurs in neurons. This spectrum of mtDNA variation had been unexplored by previous studies due to insufficient sequencing depth limiting their lower detection threshold to 1% heteroplasmy. Furthermore, by selecting our amplification and sequencing target outside of the commonly deleted mtDNA region, our sequence data reflects the entire mtDNA population of each neuron, comprising both structurally intact and deleted molecules.

In conclusion, our findings have important implications for therapy-oriented research. Pharmacological upregulation of mtDNA levels may rescue neuronal respiration and exert a neuroprotective effect in PD. Administration of agents promoting mitochondrial biogenesis via the peroxisome proliferator-activated receptor gamma pathway have indeed been associated with a significantly lower risk for PD (ref. 23). We believe that this effect may be at least partly due to compensation for somatic mtDNA damage. While further studies are needed to elucidate the molecular mechanisms underlying mtDNA copy-number regulation in aging and neurodegeneration, we believe that these may be exploited to design novel neuroprotective therapies.

## Methods

**Subject cohort.** Tissue was obtained from the Park-West study, a prospective population-based cohort which has been longitudinally followed-up for 9 years and has been described in detail[12]. All included subjects in the PD group ($n = 10$) fulfilled the National Institute of Neurological Disorders and Stroke[24] and the U.K. PD Society Brain Bank[25] diagnostic criteria for PD at their final visit. Whole-exome sequencing was performed on all individuals with PD using Roche-NimbleGen Sequence Capture EZ Exome v3 kit and paired-end 100nt sequencing on the Illumina HiSeq platform (unpublished material). To exclude monogenic effects which could confound our study, genes associated with familial parkinsonism (SNCA, LRRK2, VPS35, EIF4G1, DCTN1, CHCHD2, PARK2, PINK1, PARK7, ATP13A2, PLA2G6, FBXO7, COQ2 and SLC6A3) were extracted and screened with no evidence of known pathogenic mutations (data available on request). In addition, coverage analysis of the SNCA exons showed no evidence of copy-number variation. Controls had no known neurological disease and were matched for age ($t$ test; $P = 0.16$) and gender (Supplementary Table 1). Individuals with PD and controls had no history of mitochondrial disorder or use of antiretroviral medication which may affect mtDNA homeostasis. A complete list of the medication used by the individuals with PD during at least the last 12 months before death is shown in Supplementary Table 2.

**Neuropathological examination.** Brain tissue from individuals with PD and controls was collected at autopsy. The brain was split sagittally along the corpus callosum. One hemisphere was fixed whole in formaldehyde and the other coronally sectioned in 1 cm slices which were individually snap-frozen in liquid nitrogen. There was no significant difference in post-mortem or fixation time between individuals with PD and controls. Neuropathological examination was performed on the following brain areas from all samples of individuals with PD and controls who were above the age of 40 years: frontal cortex, hippocampus, striatum (nucleus lentiformis) cerebellum and mesencephalon (containing the substantia nigra) at the level of the third cranial nerve and pons at the level of the locus coeruleus. Serial sections (3 μm) of formalin-fixed, paraffin embedded tissue were stained with haematoxylin and eosin using a standard protocol and immunohistochemistry with antibodies against alpha-synuclein (Leica, NCL-L-ASYN, dilution 1:20), tau (DAKO, A0024, dilution 1:1,000), β-amyloid (DAKO, M0872, dilution 1:50), PGC-1α (Abcam, ab54481, dilution 1:100) and the mitochondrial membrane marker porin (Abcam, ab14734, dilution 1:10,000). Staining was performed in an automated stainer (Ventana Benchmark Ultra Stainer) and visualization using Ventana 760-500 Ultra View Universal DAB detection kit. Antibodies were diluted in Ventana Antibody Diluent (Ventana 251-018). Porin and PGC-1α staining were performed manually with the antibody diluted in TBS containing 0.1% Tween-20. MACH4 Universal HRP-polymer (Biocare M4U534) and DAB chromogen kit (Biocare DB801) were used for visualization.

Tissue morphology and staining of neurodegenerative markers were assessed by an experienced neuropathologist. Staining and assessment of porin and PGC-1α were blinded for disease status. These were initially assessed visually in situ by two researchers who were highly experienced in mitochondrial staining and histopathology (C.T. and I.F.). Subsequently, we performed a more standardized assessment independent of surrounding tissue morphology, which may affect contrast perception and suggest disease status. High-resolution microphotographs were taken at ×400 magnification using a light microscope (Leica) equipped with a Zeiss AxioCam MRc5 camera and Zen 2011 software (Carl Zeiss MicroImaging GmbH, Jena, Germany) under standardized conditions. Randomly picked dopaminergic single neurons of the substantia nigra pars compacta were cut from each photograph and stacked on a uniform black background using Fiji v2.0.0 (ref. 26). Due to the presence of neuromelanin, interfering with automated measurement procedures, the photomontages were evaluated visually by the same researchers (Fig. 6). Pathology findings are summarized in Supplementary Table 1.

**Single-neuron laser microdissection (LMD) and lysis.** For the LMD, 20 μm-thick sections were cut from frozen (−80 °C) blocks of substantia nigra, frontal cortex and cerebellum using a cryostat (Leica CM 1950) and mounted on membrane slides 1.0 PEN (Zeiss). Tissue sections were air-dried for 1 h, stained with cresyl violet (0.25% in ddH2O) for 10 min and dehydrated in graded ethanol series (75, 90 and 100%). LMD was performed on a PALM Laser microdissection

microscope (Zeiss). In the substantia nigra, the pars compacta was identified based on localization, morphology and neuromelanin pigmentation. Neuromelanin positive (dopaminergic) neurons were collected from the ventrolateral tier (A9 area), which is consistently and severely affected by neurodegeneration in PD. Pyramidal neurons and Purkinje cells were collected from the frontal cortex and cerebellar cortex, respectively. Following positive identification, neurons were measured for surface area and collected individually, one cell per tube.

A total of 871 single neurons were collected from the three brain areas of 10 individuals with PD and 21 controls. Of these, 786 samples successfully passed quality control and were used in the analyses. Analysis of neuronal surface area showed no significant difference between PD and control groups for each type of neuron. Collected cells were individually lysed in 15 μl lysis buffer (50 mM Tris pH 7.4, 0.5% Tween-20, 200 μg ml$^{-1}$ proteinase K) overnight at 56 °C, centrifuged (5 min, 10,000 r.p.m., 4 °C), incubated for 10 min at 95 °C to inactivate proteinase K, and centrifuged again. Each cell lysate was subsequently divided and used in downstream mtDNA analyses comprising determination of total copy-number, fraction of molecules harbouring deletions and ultra-deep sequencing for assessing the load of point mutations. In this way, all three types of mtDNA changes were determined in the same single neurons.

**mtDNA copy number and deletion analysis.** Total mtDNA copy number and the fraction of major arc deletion were determined in 8–10 single neurons per brain region from each of 10 individuals with PD and 21 controls: substantia nigra pars compacta (PD, $n = 74$; control: $n = 147$), frontal cortex (PD, $n = 73$; control, $n = 188$) and cerebellar Purkinje cells (PD, $n = 76$; control, $n = 150$). Our assay detected specifically deletions of the mtDNA major arc, that is, the region between the origins of replication for the light and heavy strand (Fig. 6a), where the vast majority of single and nearly all multiple deletions are known to occur[27]. Copy number and deletion analysis were performed simultaneously in the same neuron, using a duplex real-time PCR assay to detect a commonly deleted (MTND4) and rarely deleted (MTND1) targets on the mitochondrial genome[27,28]. The following primers, probes and conditions were used. MTND1: forward primer: 5′-CCCTAAA ACCCGCCACATCT-3′, reverse primer: 5′- GAGCGATGGTGAGAGCTAA GGT-3′, TaqMan MGB probe: 5′-FAM-CCATCACCCTCTACATCACCGCCC-3′. MTND4: forward primer: 5′-CCATTCTCCTCCTATCCCTCAAC-3′, reverse primer: 5′-CACAATCTGATGTTTTGGTTAAACTATATTT-3′, TaqMan MGB probe: 5′-VIC-CCGACATCATTACCGGGTTTTCCTCTTG-3′. Cell lysate (2 μl) were used per quantitative PCR (qPCR) reaction and all samples were run in triplicate. Amplification was performed on a 7500 fast sequence detection system (Life Sciences) using TaqMan Fast Advanced Master Mix (Thermofisher). Thermal cycling consisted of one cycle at 95 °C for 20 s and 40 cycles at 95 °C for 3 s and 60 °C for 30 s.

For absolute quantification, target amplification was compared with a standard curve made from a serial dilution containing equal amounts of PCR-generated and purified full-length MTND1 and MTND4 template ($10^2$–$10^6$ copies per μl). MTND1 is rarely deleted and reflects total mtDNA copy number whereas MTND4 only amplifies molecules not harbouring major arc deletions. The percentage of deletion was calculated from the difference of the two as previously described[27]. Genomic blood-DNA from two healthy individuals was used as an internal control in each experiment.

**Single-neuron ultra-deep mtDNA sequencing.** To assess mtDNA SNVs down to very-low heteroplasmy levels, we performed PCR amplification and ultra-deep sequencing of a 4,767 bp mtDNA fragment (rCRS 1157-5924) spanning two rRNA (MT-RNR1, MT-RNR2), 10 tRNA (MT-TV, L1, I, Q, M, W, A, N, C, Y) and two peptide (MT-ND1, MT-ND2) genes (Figs 7a,c and 8c). mtDNA sequencing was performed in 184 individual dopaminergic substantia nigra neurons, comprising 8–10 neurons from each of the individuals with PD ($n = 10$) and age-matched controls ($n = 10$). The presence of high levels of mtDNA deletions could introduce substantial selection bias in the amplification process towards specific mtDNA populations. mtDNA molecules harbouring deletions between the primer binding sites would be selectively overamplified due to their small size and therefore overrepresented in the sequencing results. Conversely, deletions spanning at least one primer binding site would prevent amplification of these molecules skewing towards non-deleted populations. To avoid this potentially substantial selection bias, we chose the size and location of the amplicon to be away from the major arc region where the vast majority of large deletions occur. Thus our sequencing data are representative for the entire mtDNA population in each neuron tested.

Amplification of each sample was performed in a 50 μl reaction using 5 μl of cell lysate as template, forward primer 5′-TTAAAACTCAAAGGACCTGGC-3′ and reverse primer 5′-GACCTAGTCAACGGTCGGCGAAC-3′. To minimize PCR-introduced sequence variation, we used Prime-STAR GXL DNA polymerase (reported error rate $= 1 \times 10^{-5}$, Takara Bio). Thermal cycling comprised one cycle at 92 °C for 2 min and 35 cycles of 92 °C for 10 s, 63 °C for 30 s and 68 °C for 8 min. PCR products were quality controlled by agarose gel electrophoresis using 5 μl sample and quantitated via picogreen fluorometric assay using 2 μl sample at 1:100 dilution. Samples showed concentrations in the range of 9.1–72.4 ng μl$^{-1}$. Subsequently, samples were normalized in 50 μl of nuclease-free water using 0.30–1.00 μg of stock DNA material and sonicated on a Covaris LE220 (Covaris) to a target insert size of ~400 bp. Libraries were prepared using the GSL standard whole-genome library prep protocol with 1.8 × clean-up to retain all fragments ≤400 bp insert size and quality controlled by qPCR (KA PA SYBR FAST qPCR, Kapa Biosystems). Products were sequenced using paired-end 125nt sequencing on the Illumina HiSeq 2500, v4 chemistry and generating ~1 Gb of data per sample. Sequencing was performed at HudsonAlpha Institute for Biotechnology (Huntsville, AL).

**Sequence data analysis.** Illumina paired-end reads in FASTQ format were mapped to the rCRS reference sequence using BWA aligner v6.2 (ref. 29). From the resulting alignments, only bases with a PHRED-score ≥30 and a mapping quality of ≥30 were used for the subsequent analyses. SAMtools version 1.3-8 (ref. 30) was used to sort and index the alignments and to remove duplicates. Pileup files were generated for each of the samples using the mpileup utility of SAMtools. Subsequently, pileup files were parsed to calculate the relative frequencies of each nucleotide per mtDNA position. Variants were called requiring a minimum coverage of 30 reads for the variant base and the lowest limit for heteroplasmic detection was set to 0.1%. mtDNA haplotype were defined using the mt-classifier script (v.0.2; ref. 31) and requiring HF ≥80% for the variants defining the haplotype in each neuron. Samples were quality controlled for haplotype consistency across all neurons of an individual. Samples with inconsistent haplotypes were discarded from the analyses as they were likely to be contaminated. A total of 144 samples (144 neurons from 10 individuals with PD and 10 controls) passed quality filters and were used for the analyses of mutational burden. For the statistical analyses, variants were categorized in three groups according to HF: low-frequency heteroplasmy (HF 0.1–1%), high-frequency heteroplasmy (HF 1–98%) and homoplasmy (98–100%).

**Statistical analyses.** Statistical analyses were performed in SPSS (v.20.0.0.1), Graph Pad Prism (v.6) and R (v.3.2.3). Group comparisons for mtDNA deletion and copy number were done by Mann–Whitney U test. Comparison/correlation between different types of mtDNA changes and/or age was done by linear regression analysis. To assess whether age or level of mtDNA deletion was a stronger determinant of mtDNA quantity in a neuron, we performed two independent single linear regression analyses using age or copy number as predictors. To quantify the net association of each of the predictors with the outcome variable we performed a multiple linear regression model. Comparison of point mutational load between groups was done by $\chi^2$ test. Mutational burden distribution between different mtDNA genes in individuals with PD and controls was compared by ANOVA, followed by group comparisons by $t$ tests and Bonferroni correction for multiple testing. Investigators were blinded to the state of samples (PD or control) for all mtDNA studies. Tissue sections from which the neurons were microdissected were identified by unique numbers only.

**Ethical considerations.** These studies were approved by the Regional Committee for Medical and Health Research Ethics, Western Norway (REK 131/04 and REK 2010/23). Informed consent was obtained from all subjects.

**Data availability.** The pipeline for mtDNA sequence analysis was implemented using a script in Python 2.7 which is available on request. The data sets generated and/or analysed during the current study are available from the corresponding author on reasonable request.

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

## Acknowledgements

We thank the patients and their families for contributing to this study. We are grateful to Dr Brage Brakedal for the discussions of the statistics. This work was supported by grants from the Regional Health Authority of Western Norway (grant no 911903 and 911988) and the Research Council of Norway (grant no 240369/F20).

## Author contributions

C.T. conceived, designed, directed and funded the study. C.D. and C.T. drafted the manuscript. C.D. and I.F. collected the single neurons and performed the mtDNA analyses. G.S.N. performed the analyses of the deep-sequencing data. C.D., I.F. and N.O. performed the neuropathological staining. H.M., C.T. and I.F. performed the neuro-pathological assessment. S.K. and P.K.L. contributed control material and performed neuropathological examinations. J.P.L. and O.-B.T. provided access to the patient material and performed the clinical assessment of the patients. K.H. and L.A.B partici-pated in interpreting the data and critically revising the manuscript. All authors con-tributed to the final version of the manuscript.

## Additional information

**Competing financial interests:** The authors declare no competing financial interests.

**Reprints and permission** inform is available online at http://npg.nature.com/reprintsandpermissions/

