## [Peer Review File · Nature Communications]

Reviewers' comments:

Reviewer #1 (Remarks to the Author):

NCOMMS-16-07688-T

Defective mitochondrial DNA homeostasis in the dopaminergic substantia nigra of patients with Parkinson's disease" by Dr Tzoulis and colleagues.

This is a nice brief communication in which the authors show that impact of mitochondrial DNA deletions in PD substantia nigra neurons is mediated through a failure to compensate by increasing mtDNA copy number.

It would seem that the authors' findings are consistent with a mitochondrial biogenesis defect and authors should consider expanding their discussion to highlight this point. Otherwise I have no major concern.

Reviewer #2 (Remarks to the Author):

The manuscript by Dolle and colleagues analyzes the levels of mtDNA deletions and the mtDNA content in dissected single neurons from postmortem brain of normal individual at different ages and in PD brains. They find that large-scale deletions accumulate with age in SN DA neurons, but not in cortical or cerebellar neurons. They also determine that the amount of mtDNA increases proportionally with the increase of deleted genomes in control individuals, but not in PD subjects, suggesting that the latter fail to compensate for the accumulating deletions by increasing the pool of normal mtDNA. They suggest that this failure to increase mtDNA content is responsible for a bioenergetic failure in DA neurons of PD patients.

The work confirms previous reports of age-related increase of somatic mtDNA deletions with aging and highlights the potential role of such deletions in PD pathogenesis. The most novel aspect of this manuscript lies in the quantification of total mtDNA, which, combined with the increasing deletions, provides a ratio of normal:deleted mtDNA that is unfavorable in PD subjects. The work is technically well executed. The number of subjects analyzed in both the normal aging group and in the PD group, which is well matched with controls by age and gender and the number of neurons analyzed is quite impressive. In addition to deletions they also analyzed point mutations, which did not appear to differ in PD and controls, further reaffirming the preferential expansion of the pool of deleted mtDNA. Nevertheless, the work does not directly address the causes of deletion accumulation in aging and particularly in PD. Also, they do not address the mechanisms leading to mtDNA increase associated with accumulating deletions and the reasons for the failure of mtDNA proliferation in PD.

Specific points:

- 1) The work is of high technical quality and an important addition to the growing evidence for mitochondrial abnormalities in aging and in PD, but does not provide new mechanistic insights. Naturally, this is a limitation of work performed on post-mortem samples that can hardly be overcome.
- 2) It would be meaningful to investigate the levels of known components of the signaling pathways that regulate mitochondrial biogenesis and specifically mtDNA replication. It is unclear if this is feasible in single neurons dissected from the brain slices.
- 3) There is no analysis of the consequences of higher deletions in the surviving SN neurons in aging and PD brains. It would be expected that respiratory chain defects would also directly correlate with mtDNA deletion and inversely correlate with normal mtDNA. This type of functional analysis was not performed.
- 4) Ultimately, it is difficult to firmly establish that a demise of oxidative phosphorylation is caused by the failure to compensate for accumulating deletions and that it is the cause of DA neurons

loss.

Reviewer #3 (Remarks to the Author):

In this manuscript, Dölle et al., investigated the role of mitochondrial DNA copy number in aging and in neurodegeneration of dopamine neurons of the SNc. First, they characterized mtDNA changes in different populations of neurons from control brain samples at different ages. Then, they investigated the integrity of mtDNA in individual neurons from patients with sporadic PD and compared their results to ages and sex-matched controls. Finally, they verified whether point mutations also contribute to neuronal mtDNA pathology in PD by performing deep sequencing analysis on a fragment of mtDNA. They reported that the main determinants of somatic neuronal mtDNA damage in human are deletions and copy number regulation but not point mutations. One of the most important findings is that they report that SNc neurons can compensate the somatic deletion by a regulation of mtDNA copy number and this compensation seems blunted in SNc neurons from PD brains.

This is an interesting study that raises important information about the possible mechanisms contributing to PD. One weakness however is that a recent study (Coxhead et al., 2016) showed that somatic mtDNA mutations are an important component of PD. This is in contradiction with the present study.

Major points:

I understand that this is a challenging experiment but their results on single neuron sequencing are only based on 8 dopaminergic neurons per brain sample from 5 PD patients and 5 controls. To be more convincing, they need to provide a more broad analysis on more neurons from more PD and control samples.

The authors seemed to confuse correlation and causality in the interpretation of their results, especially in their characterization of age-dependent mtDNA changes. The design of the study does not allow drawing conclusions about the up-regulation of mtDNA copy number according to age or deletion. The authors showed only an association between mtDNA copy with age and mtDNA deletion. Also, although their correlations are significant, some are modest and caution should be made when drawing conclusions.

In the first part of the study, authors show a correlation between age and deletion among the control. In figure 2b, they present a significant difference between the PD and control regarding the wildtype mtDNA copy number. How does the difference in age between the PD and control groups contribute to the results obtained? When looking at the 10 oldest controls, there is still a mean difference of about 6 years between the groups, the PD group being older than control.

For ultra deep sequencing, the authors should provide justification about the fragment chosen and which genes were included. Figure S5a is incomplete. Information about gene specific SNVs should be provided.

Considering that there is between 2 and 10 mtDNA copies per mitochondrion and several mitochondria per neuron, it might have been of great interest to report mtDNA copy number per mitochondrion. This could help to decipher between replicative/transcription of mtDNA or mitochondrial biogenesis.

Regarding mtDNA copy number and deletion analysis, they analyzed 8-10 single neurons per brain of 10 PD and 21 controls. It is reasonable to think that most vulnerable neurons are already degenerated in these samples and the remaining neuromelanin neurons might represent a

subpopulation of DA neurons. The authors should describe or provide a map of where in the SNc they isolated these neurons in both PD and control samples. Without sampling approximately the same subpopulation, the results are difficult to interpret.

Overall the manuscript is well written but sometimes the manuscript is suffering from the lack of crucial information that could be problematic for a non-expert in the field. I-e, they should better define "major arc deletion", "homoplasmic and heteroplasmic", "transition to transversion ratio"...

Minor point:

The authors quantified MTND4 and MTND1 and calculated the percent of deletion from the difference of the two. Because MTND1 can also be deleted, they should justify why they have not made their measurements as a relative abundance of MTND4 and MTND1 to a nuclear encoded gene and/or used additional well-preserved regions.

They should add in supplemental files the medication the patients received (and the controls if applicable) as drug treatments might affect mtDNA copy number (Feng, 2013).

Reviewer 1

This is a nice brief communication in which the authors show that impact of mitochondrial DNA deletions in PD substantia nigra neurons is mediated through a failure to compensate by increasing mtDNA copy number.

Comment

It would seem that the authors findings are consistent with a mitochondrial biogenesis defect and authors should consider expanding their discussion to highlight this point. Otherwise I have no major concern.

Author Reply

We thank the reviewer for evaluating our work and for the comments. This is indeed a very relevant point which we have further addressed and discussed to the best of our ability within the time confines for revision. As requested also by reviewer-2, we have performed supplementary experiments looking at markers of overall mitochondrial mass and biogenesis. The new results have been included in the manuscript and we have expanded our discussion accordingly (Please see: Figure 5; Supplementary Figure 1; Results page 5, lines 119-129; Discussion page 8, lines 179-193, Methods page 12-13, lines 261-264 and 166-279).

Reviewer 2

Comment-1

1) The work is of high technical quality and an important addition to the growing evidence for mitochondrial abnormalities in aging and in PD, but does not provide new mechanistic insights. Naturally, this is a limitation of work performed on post-mortem samples that can hardly be overcome.

Author Reply-1

We thank the reviewer for the evaluation of our work and fully agree that there is a need to elucidate the mechanism(s) underlying the apparently impaired mtDNA maintenance in PD. In the absence of good cell or animal model reflecting sporadic PD in humans, patient tissue remains one of the most accurate media in which to study the disease. Unfortunately, as the reviewer rightfully points out, the possibility to investigate mechanisms in *post-mortem* brain tissue is limited. In order to increase mechanistic insight, we have performed additional experiments to the best of our ability within the time confines for revision. We investigated markers of mitochondrial mass and biogenesis *in situ* and included these findings to our results and discussion. Please see more details in our answer to comment-2 below.

Comment-2

2) It would be meaningful to investigate the levels of known components of the signaling pathways that regulate mitochondrial biogenesis and specifically mtDNA replication. It is unclear if this is feasible in single neurons dissected from the brain slices.

Author Reply-2

We thank the reviewer for the valuable suggestion. We have now performed supplementary experiments looking at markers of overall mitochondrial mass and biogenesis in our tissue samples by immunohistochemistry. Although moderately sensitive, this technique allows us to assess the markers *in situ* and in individual neurons as opposed to RNA-expression experiments or protein blots which would be severely confounded by the altered cellular composition in the substantia nigra of PD (relating to this problem, please see also under comments of Reviewer-3, Author Reply-1, point-1 “Homogenate vs. single cells”). Although immunohistochemistry is only moderately sensitive, it should detect major differences in neuronal mitochondrial mass or PGC-1 α expression. As no difference was detected between patients and controls however, a major impairment of mitochondrial biogenesis is unlikely. We therefore believe our findings are more consistent with a defect of mtDNA maintenance. The new results have been included in the text and we have expanded our discussion accordingly (Please see: Figure 5; Supplementary Figure 1; Results page 5, lines 119-129; Discussion page 8, lines 179-193, Methods page 12-13, lines 261-264 and 166-279).

Further studies to understand the origin and mechanism of impaired mtDNA maintenance in PD are warranted and indeed already ongoing in our lab. We hope the reviewer agrees that further elucidation of the underlying mechanism, while essential, is beyond the scope of the present study.

Comment-3

3) There is no analysis of the consequences of higher deletions in the surviving SD neurons in aging and PD brains. It would be expected that respiratory chain defects would also directly correlate with mtDNA deletion and inversely correlate with normal mtDNA. This type of functional analysis was not performed.

Author Reply-3

We agree that this analysis would be relevant. In fact, we performed numerous experiments attempting to correlate mtDNA deletion with respiratory defects in single neurons already for the original version of this manuscript. To this end, we used immunohistochemistry for respiratory chain complexes or COX/SDH histochemistry, combined with subsequent mtDNA analysis in single neurons. Unfortunately, mtDNA could not be accurately determined in these samples due to technical limitations. Specifically, qPCR was severely influenced by the presence of 3,3'-diaminobenzidine (DAB) residue, used in both immunohistochemistry and histochemistry to visualize the complexes. This has been observed before (PMID: 22647770). In fact, the interfering effect of DAB staining on PCR was further verified and studied by more experiments and a method-related manuscript describing the issue is in preparation. Alternative staining methods using immunofluorescence were also tried. These were however, substantially less sensitive than DAB for identifying respiratory deficient neurons in the thick, frozen brain sections required for microdissection and mtDNA analyses. Moreover, qPCR results from fluorescently stained sections also showed larger than usual variation, suggesting the fluorescent dyes may also influence the qPCR. Therefore, we are unfortunately unable to present additional data correlating mtDNA deletion levels and respiratory chain defects in the same neuron at this point. We hope that the reviewer understands this technical limitation.

Comment-4

4) Ultimately, it is difficult to firmly establish that a demise of oxidative phosphorylation is caused by the failure to compensate for accumulating deletions and that it is the cause of DA neurons loss.

Author Reply-4

We agree with the reviewer and refer also to our answer to the previous comment with regard to technical feasibility. We have also modified this point in our discussion and toned down this argument (Please see: Discussion page 8, lines 174-176).

Reviewer 3

Comment-1

This is an interesting study that raises important information about the possible mechanisms contributing to PD. One weakness however is that a recent study (Coxhead et al., 2016) showed that somatic mtDNA mutations are important component of PD. This is in contradiction with the present study.

Author reply-1

We thank the reviewer for the thorough evaluation of our work. We are indeed aware of the study by Coxhead *et al.* and had in fact included this in our discussion. We respectfully disagree however that this difference reflects a weakness with our work. In fact, the two studies employ very different strategies and have substantially different sensitivities. Coxhead et al., measure mtDNA mutations in a mixture of cells from homogenized tissue, preferentially amplify non-deleted mtDNA and lack sensitivity for very low-level heteroplasmy (< 1%) due to lower sequencing depth. We measure mtDNA in the target cell of PD (i.e. dopaminergic neurons of the pars compacta of the substantia nigra), assess the entire mtDNA population independent of deletions and confidently assess very low-level heteroplasmy (< 1%), where the vast majority (> 95 %) of changes occur. We discuss these differences in more detail point-by-point below:

1) Homogenate vs. single cells

The study by Coxhead et al., was performed in tissue homogenate and not individual neurons. It is therefore likely to be confounded by the different cellular composition of the *substantia nigra* between PD and control, rather than reflect true changes in neuronal mtDNA. The *substantia nigra pars compacta* of patients with PD shows at least ~ 60 % loss of the dopaminergic neurons, which are the principal source of mtDNA in the region, and proliferation of glial cells with less mtDNA and a drastically different biology. Therefore, any experiment in homogenate is likely to be severely biased by cellular composition in the sample.

2) MtDNA deletions can skew the amplification and sequencing towards

Strategies employed in the mtDNA amplification prior to sequencing greatly influence whether all or only part of the mtDNA population is captured. In the study by Coxhead et al., mtDNA was amplified in two fragments, both of which overlap with the deletion-prone major arc. This would lead to selective amplification of mtDNA molecules with deletions encompassed by the primer binding sites, whereas molecules with deletions spanning primer sites would be excluded from amplification. Given that ~ 40 %, in average, of mtDNA molecules per neurons harbor major arc deletions, this would introduce a substantial selection bias and sequence data would not be representative for the entire mtDNA population. By selecting our sequencing target outside of the major arc region, we are able to circumvent this limitation and generate sequence data that reflect the entire mtDNA population of each neuron. We therefore believe that our mtDNA sequencing data are highly precise and accurately reflect the somatic mutational state

3) Depth of sequencing

Our sequencing depth is substantially higher than that employed by Coxhead et al. We are therefore able to confidently detect variants down to a frequency of 0.1 % (vs. 1% in Coxhead et al), even using very stringent filtering criteria such as 30 minimum supporting reads (vs. 10 in Coxhead et al). With this sensitivity, we are able to show that vast majority (> 95 %) of all somatic mtDNA point mutations in neurons occurs at heteroplasmic frequencies below 1%. This spectrum of mtDNA variation had been unexplored by previous studies, including Coxhead et al, due to insufficient sequencing depth.

We therefore believe that our study addresses the question of somatic point mutation in PD in a highly accurate and specific manner. We do agree with the reviewer that the discrepancy from the earlier study by Coxhead et al., should be discussed and have included this in our discussion (Please see: Discussion page 9, lines 195-214).

Comment-2

I understand that this is a challenging experiment but their results on single neuron sequencing are only based on 8 dopaminergic neurons per brain samples from 5 PD patients and 5 controls. To be more convincing, they need to provide a more broad analysis on more neurons from more PD and control samples.

Author reply-2

We have now more than doubled our sample to a total of 184 neurons from all 10 PD patients and 10 controls. The results of the analyses in the expanded sample are highly consistent with those of the initial experiment. This confirms reproducibility and consistency in a larger population and also across two experimental batches of microdissection, PCR and sequencing. We have adjusted the Method (page 15, lines 338-340), results (page 6, §lines 132-135), figures (Fig. 6, 7) and discussion accordingly.

This is the largest to date ultra-deep mtDNA sequencing experiment performed in the target cell of PD (i.e. single *substantia nigra* neurons). It is also the first study sensitive enough to confidently assess neuron-specific somatic mutations at heteroplasmic levels below 1 %, where we show that the vast majority (96 %) of somatic mtDNA point mutations occurs in substantia nigra neurons.

The samples come from a clinical material of unique quality comprising validated, true sporadic PD. The Park-West cohort (Alves *et al.*, 2009), is entirely population based, prospectively identified and followed-up for years by clinical experts in movement disorders. In addition all samples have been genetically and pathologically characterized. We therefore believe that our results are highly representative for true sporadic PD and hope the reviewer agrees that sample size is now satisfactory.

Comment-3

The authors seemed to confuse correlation and causality in the interpretation of their results, especially in their characterization of age-dependent mtDNA changes. The design of the study does not allow drawing conclusion about the up-regulation mtDNA copy number according to age or deletion. The authors showed only an association between mtDNA copy with age and mtDNA deletion. Also, although their correlations are significant, some are modest and caution should be made when drawing conclusion.

Author reply-3

We are aware of the differences between correlation and causation. As the reviewer rightfully points out, causation is hard to prove in a post-mortem sample. However, the relationship and dependency between variables may still be assessed. Here we corrected for the effect of age by performing a multiple regression analysis (Results, page 3, lines 73-76), which shows that only deletion is a statistically significant predictor of copy number. Although we agree that this does not infer causality, it does show that deletion levels can predict neuronal mtDNA copy number, whereas age cannot. This is exactly what we are saying in the text, i.e. that deletion, but not age, is a statistically significant predictor of neuronal copy number.

Our positive correlations between deletion and copy number in the *substantia nigra* are highly significant and the strength of the association is considerable. We do agree with the reviewer however in that caution should be exercised when interpreting associations in a context of biological mechanisms and have moderated this in the manuscript (Results, page 4, lines 81-82).

To make the interpretation of the correlations more reader-friendly and include the direction of the relationship between variables, we now give in the text the linear correlation coefficient (**r**) values, instead of R^2 (or both where appropriate).

Comment-4

In the first part of the study, authors show a correlation between age and deletion among the control. In the figure 2b, they present a significant difference between the PD and control regarding the wildtype mtDNA copy number. How the difference in the age between the PD and control groups contributes to the results obtained? When looking at the 10 oldest controls, there is still a mean difference of about 6 years between the groups, PD group being older than control.

Author reply-4

The age difference between PD patients and controls is not statistically significant (t-test $P = 0.16$) and should therefore have no significant impact on the results. This information has been added to the manuscript (Methods, page 11, line 240).

To confirm that age had no effect, we repeated the analysis including only exactly age-matched individuals (PD $n = 8$, mean age 79.8 ± 6.5 years; Controls $n = 8$, mean age $80.1 \pm$

8.4 years). These analyses gives identical results to the original, showing no difference for total mtDNA copy number ($P = 0.4$) and a highly significant difference for wild-type mtDNA levels (PD mean $17,404 \pm 8,888$, controls mean $20,618 \pm 7,753$ copies / neuron, $P = 0.0052$).

Comment-5

For ultra deep sequencing, the authors should provide justification about the fragment chosen and which genes were included. The figure s5a is incomplete. Information about gene specific SNV should be provided.

Author reply-5

The mtDNA region was chosen because this portion of the mtDNA is commonly spared by deletions, most of which occur in the major arc. As explained in detail in our reply to “Comment-1” this is of strategic importance for avoid deletion-induced bias in amplification and sequencing. Molecules with deletions spanning primer binding sites would not amplify at all, whereas molecules with deletions between primer sites would be smaller and selectively overamplify resulting in overrepresentation in the sequencing (please see also reply to Comment-1, point-2).

We have improved the figure (see figure 6a and 6b) and explain this in detail in the method, including a list of sequenced genes (Methods, page 15, lines 334-348; Figure 6a,c and Figure 7c). Information about gene specific SNV has been provided in the Supplementary Table-3. As we found no difference in the mutational load between PD patients and controls, we felt that descriptive details of specific variants would be unnecessarily lengthy without adding scientific value. If the editor deems it necessary, we will be happy to submit complete lists of variants along with functional annotations as supplementary material.

Comment-6

Considering that there is between 2 and 10 mtDNA copy per mitochondrion and several mitochondria per neurons, it might have been of great interest to report mtDNA copy number per mitochondrion. This could help to decipher between replicative/transcription of mtDNA or mitochondrial biogenesis.

Author reply-6

We agree that this information would be interesting, but admit that it is not clear to us how estimating mtDNA copies per mitochondrion would be at all possible in this work. How would we estimate the absolute number of “single” mitochondria in each microdissected neuron in order to establish a ratio of mtDNA molecules per mitochondrion?

That being said, we agree completely that it would be interesting to distinguish between mtDNA homeostasis and overall mitochondrial biogenesis. To this end we have supplemented our studies with immunohistochemical analyses of markers for mitochondrial mass and biogenesis. These experiments show no detectable difference between PD patients and controls, suggesting no major defect in mitochondrial biogenesis. We would also like to refer

to our replies to Reviewer-2, Comment-2 for more details. The new results have been included in the text and we have expanded our discussion accordingly (Please see: Figure 5; Supplementary Figure 1; Results page 5, lines 119-129; Discussion page 8, lines 179-193, Methods page 12-13, lines 261-264 and 166-279).

Comment-7

Regarding mtDNA copy number and deletion analysis, they analyzed 8-10 single neurons per brain of 10 PD and 21 controls. It is reasonable to think that most vulnerable neurons are already degenerated in these samples and the remaining neuromelanin neurons might represent a subpopulation of DA neurons. The authors should describe or provide a map of where in the SNc they isolated these neurons in both PD and control samples. Without sampling approximately the same subpopulation, the results are difficult to interpret.

Author reply-7

We agree with this important point. All neurons were picked from exactly the same area of patients and controls: the substantia nigra pars compacta, ventrolateral tier (A9 region). Only neurons positively identified as dopaminergic based on neuromelanin content were collected. We have added details in the main text (Results page 3, line 67; Methods page 13, lines 186-290), and shown a depiction of representative sections used for microdissection in Figure-1.

Comment-8

Overall the manuscript is well written but sometimes the manuscript is suffering from the lack of crucial information that could be problematic for a non-expert in the field. I.e., they should better define "major arc deletion", "homoplasmic and heteroplasmic", "transition to transversion ratio"...

Author reply-8

We have added in the text explanations for technical terms including "major arc" (Figure 6a, Method page 14, lines 309-311), "homoplasmic / heteroplasmic", "transition" and "transversion" (Results page 6, lines 144-146).

Minor point:

Comment-9

The authors quantified MTND4 and MTND1 and calculated the percent of deletion from the difference of the two. Because MTND1 can also be deleted, they should justify why they have not made their measurements as a relative abundance of MTND4 and MTND1 to a nuclear encoded gene and/or used additional well-preserved regions.

Author reply-9

As our analyses were done in **single** microdissected neurons, it was not technically possible to include a nuclear gene in the equation due to the extremely small amount of nuclear DNA. Moreover, unlike studies in tissue homogenates where a nuclear encoded gene is necessary in

order to correct for the number of cells in the sample, we accurately measured the absolute copy number of mtDNA per neuron by comparing sample amplification to that of a carefully titrated standard curve. This is a validated method and one that has been successfully used in numerous studies (see He et al., 2002, PMID: 12136116, Bender et al., PMID: 16604074).

While it is true that *MT-ND1* can be deleted, this is rare and quantitatively negligible in comparison to deletions of the major arc where *MT-ND4* is located. In fact, thorough mapping of the mtDNA deletion breakpoints shows that *MT-ND1* is intact in the vast majority (94 %) of affected in patients with single deletions and nearly all with multiple mtDNA deletions, which is the case in PD (see He et al., 2002, PMID: 12136116. Detailed overview of mapped breakpoints can be found at: “MITOMAP: A Human Mitochondrial Genome Database (2001)” <http://www.gen.emory.edu/mitomap.html>)

The *MT-ND4* to *MT-ND1* ratio is a well-established method for estimating mtDNA deletion levels in single cells. It has been used in multiple studies and was validated against Southern blot, which is considered the gold standard (see He et al., 2002, PMID: 12136116, and Krishnan et al., 2007, PMID: 17662684). We therefore consider our assessment of mtDNA deletion to be accurate enough for the purpose of these studies.

Comment-10

They should add in supplemental files the medication the patients received (and the controls if applicable) as drug treatments might affect mtDNA copy number (Feng, 2013).

Author reply-10

We have added this information in the supplement (Supplementary Table 3). None of the medication used by our patients is known to affect mtDNA levels.

REVIEWERS' COMMENTS:

Reviewer #2 (Remarks to the Author):

The manuscript by Dolle and colleagues has been revised to include a larger number of samples and to test markers of mitochondrial biogenesis. The text has been modified to reflect some of the reviewers' comments. The work is well done and potentially interesting for a broad audience. There are some points that should be addressed to further improve the manuscript.

- 1) Line 84: What is the number of WT mtDNAs per cell in aged DA neurons? In other words, is the total mtDNA increase sufficient to generate a WT mtDNA amount comparable to younger DA neurons?
- 2) Line 131: the paragraph title should be modified: it should state that no difference in somatic point mutations levels was found. None of the findings necessarily imply pathogenicity or lack of thereof.
- 3) Line 171: The statement on compensation should be toned down, since it is unknown whether there is functional loss and/or compensation.
- 4) Line 176: The findings presented here suggest a decline in relative proportion of WT mtDNA not a depletion of total mtDNA. Thus, the similarity made with the referenced work is somehow inaccurate.
- 5) Line 194: Potential causes for failure to upregulate mtDNA should be discussed, since they suggest that this is the root cause of the problem.
- 6) Line 198: The sentence should be rephrased as "changes in DA neurons".

Reviewer #3 (Remarks to the Author):

The authors now highlight the difference and strength of their study compared to Coxhead et al 2016 and included them in the discussion. The authors have now more than doubled their initial sample to a total of 184 neurons from all 10 PD patients. We appreciate this increase in sample size which renders the values presented in the study much more convincing. It was essential. The authors made an effort to make the interpretation of the correlations more reader-friendly and slightly moderated the interpretation of the associations in a context of biological mechanisms.

The manuscript now includes the information about the age difference between PD patients and controls that was not statistically significant and to confirm that age had no effect, they repeated the analysis including only exactly age-matched individuals.

The figure (see figure 6a and 6b) was indeed improved.

The overall changes are satisfactory and the manuscript should be published.

Reviewer 2

The manuscript by Dolle and colleagues has been revised to include a larger number of samples and to test markers of mitochondrial biogenesis. The text has been modified to reflect some of the reviewers' comments. The work is well done and potentially interesting for a broad audience. There are some points that should be addressed to further improve the manuscript.

1) Line 84: What is the number of WT mtDNAs per cell in aged DA neurons? In other words, is the total mtDNA increase sufficient to generate a WT mtDNA amount comparable to younger DA neurons?

Author Reply

We thank the reviewer for this clarifying comment. The total mtDNA increase is indeed sufficient to maintain the WT mtDNA pool in aging substantia nigra neurons. We have added this information to the manuscript (pages 4-5, marked in text)

2) Line 131: the paragraph title should be modified: it should state that no difference in somatic point mutations levels was found. None of the findings necessarily imply pathogenicity or lack of thereof.

Author Reply

We have rephrased the title of the paragraph and removed reference to pathogenicity as requested by the reviewer. We would like to emphasize however that our detailed mtDNA analyses by ultra-deep sequencing show no quantitative **or** qualitative differences in the mtDNA point mutational load of PD patients and controls. The fact that the mtDNA point mutational load of persons with PD is identical to that of healthy individuals should be sufficient to deduce lack of a specific pathogenic contribution in PD. On these grounds, we do not feel that our conclusion is unsubstantiated.

3) Line 171: The statement on compensation should be toned down, since it is unknown whether there is functional loss and/or compensation.

Author Reply

We have rephrased this statement and toned down the reference to compensation as requested.

4) Line 176: The findings presented here suggest a decline in relative proportion of WT mtDNA not a depletion of total mtDNA. Thus, the similarity made with the referenced work is somehow inaccurate.

Author Reply

We agree with the reviewer that our findings are different from those in the referenced work and have removed this reference.

5) Line 194: Potential causes for failure to upregulate mtDNA should be discussed, since they suggest that this is the root cause of the problem.

Author Reply

We have now included this in our discussion.

6) Line 198: The sentence should be rephrased as “changes in DA neurons”.

Author Reply

We have rephrased the sentence as requested.

Reviewer 3

The authors now highlight the difference and strength of their study compared to Coxhead et al 2016 and included them in the discussion. The authors have now more than doubled their initial sample to a total of 184 neurons from all 10 PD patients. We appreciate this increase in sample size which renders the values presented in the study much more convincing. It was essential. The authors made an effort to make the interpretation of the correlations more reader-friendly and slightly moderated the interpretation of the associations in a context of biological mechanisms.

The manuscript now includes the information about the age difference between PD patients and controls that was not statistically significant and to confirm that age had no effect, they repeated the analysis including only exactly age-matched individuals.

The figure (see figure 6a and 6b) was indeed improved.

The overall changes are satisfactory and the manuscript should be published.

Author Reply

We thank the reviewer for the comments. No specific further requests are made.